# Prioritization of Microorganisms Isolated from the Indian Ocean Sponge *Scopalina hapalia* Based on Metabolomic Diversity and Biological Activity for the Discovery of Natural Products

**DOI:** 10.3390/microorganisms11030697

**Published:** 2023-03-08

**Authors:** Alexandre Le Loarer, Rémy Marcellin-Gros, Laurent Dufossé, Jérôme Bignon, Michel Frédérich, Allison Ledoux, Emerson Ferreira Queiroz, Jean-Luc Wolfender, Anne Gauvin-Bialecki, Mireille Fouillaud

**Affiliations:** 1Laboratory of Chemistry and Biotechnology of Natural Products, Faculty of Science and Technology, University of La Réunion, 15 Avenue René Cassin, CS 92003, CEDEX 09, 97744 Saint-Denis, France; 2Institute of Pharmaceutical Sciences of Western Switzerland, University of Geneva, CMU-Rue Michel-Servet 1, CH-1211 Geneva, Switzerland; 3School of Pharmaceutical Sciences, University of Geneva, CMU-Rue Michel-Servet 1, CH-1211 Geneva, Switzerland; 4Institute of Chemistry of Natural Substances (ICSN), CNRS UPR 2301, Université Paris-Saclay, 1, av. de la Terrasse, CEDEX, 91198 Gif-sur-Yvette, France; 5Pharmacognosy Laboratory, Department of Pharmacy, Centre Interfacultaire de Recherche sur le Médicament (CIRM), University of Liège, Quartier Hôpital, Avenue Hippocrate 15, Bat. B36, Campus du Sart-Tilman, B-4000 Liege, Belgium

**Keywords:** natural products, marine microorganisms, specialized metabolites, molecular network, dereplication, cytotoxic activity, antiplasmodial activity, prioritization method

## Abstract

Despite considerable advances in medicine and technology, humanity still faces many deadly diseases such as cancer and malaria. In order to find appropriate treatments, the discovery of new bioactive substances is essential. Therefore, research is now turning to less frequently explored habitats with exceptional biodiversity such as the marine environment. Many studies have demonstrated the therapeutic potential of bioactive compounds from marine macro- and microorganisms. In this study, nine microbial strains isolated from an Indian Ocean sponge, *Scopalina hapalia,* were screened for their chemical potential. The isolates belong to different phyla, some of which are already known for their production of secondary metabolites, such as the actinobacteria. This article aims at describing the selection method used to identify the most promising microorganisms in the field of active metabolites production. The method is based on the combination of their biological and chemical screening, coupled with the use of bioinformatic tools. The dereplication of microbial extracts and the creation of a molecular network revealed the presence of known bioactive molecules such as staurosporin, erythromycin and chaetoglobosins. Molecular network exploration indicated the possible presence of novel compounds in clusters of interest. The biological activities targeted in the study were cytotoxicity against the HCT-116 and MDA-MB-231 cell lines and antiplasmodial activity against *Plasmodium falciparum* 3D7. *Chaetomium globosum* SH-123 and *Salinispora arenicola* SH-78 strains actually showed remarkable cytotoxic and antiplasmodial activities, while *Micromonospora fluostatini* SH-82 demonstrated promising antiplasmodial effects. The ranking of the microorganisms as a result of the different screening steps allowed the selection of a promising strain, *Micromonospora fluostatini* SH-82, as a premium candidate for the discovery of new drugs.

## 1. Introduction

Despite the considerable medical progress observed in recent decades, many human or animal pathologies remain incurable. In developed areas, a high mortality rate is often due to heart diseases and cancers [1], mainly linked to the particular lifestyles of these populations (unbalanced diet, physical inactivity, alcohol consumption or tobacco use) [2,3,4]. In developing countries, deaths are more often attributed to communicable diseases such as malaria (nearly 190 million cases in 2020 concentrated on the African continent) [5] or HIV (Human Immunodeficiency Virus) [1]. Recently, the World Health Organization (WHO) has worryingly observed increasing resistances appearing to the current therapies, for instance, *Plasmodium* parasites, responsible for malaria, becoming partially resistant to artemisinin [5] or MRSA (methicillin-resistant *Staphylococcus aureus*), which is also resistant to aminoglycosides, macrolides, tetracycline, chloramphenicol and lincosamides [6]. Additionally, the global progression of cancers and the resistance to current drug treatments [5,6,7,8] urgently require researchers to find new alternatives.

Natural products (NPs) are known to be a formidable source of drugs [9], and the discovery of novel bioactive compounds is, worldwide, a priority public health issue. As terrestrial environments have been extensively studied, the recent studies focus on less frequently explored habitats, such as marine biotopes [10]. These habitats have already shown their potential to shelter unusual organisms favorable to the discovery of bioactive compounds that can be used in the pharmaceutical industry [10,11,12,13,14].

Macroorganisms living in the marine environment, such as sponges, are already recognized as producers of a wide range of marine natural products (MNPs) with original chemical structures [15,16]. *Scopalina hapalia* is a sponge of the order Scopalinida [17]. This species is distributed in different areas such as Australian waters and the Indian Ocean (Zanzibar and Mayotte) [18,19]. It has shown interesting activities such as anti-microbial, anti-aging and cytotoxic effects [19,20,21]. However, this species is not well described for its metabolites. One article gives the chemical composition of this sponge [19], highlighting the presence of butenolide derivatives, phospholipids or even brominated compounds. These compounds can be synthetized by the macro-organisms themselves or by their associated microbial community [22,23]. Another hypothesis is that the production of bioactive metabolites is possibly due to the existence of this holobiont composed of the macroorganism and its microbiome, which together form a discrete ecological unit through symbiosis or various thin interactions.

This microbiome can be composed of different microbial populations such as fungi, bacteria or microalgae [24,25,26,27,28]. Amongst these communities, actinobacteria are widely known to produce bioactive molecules of interest [26,27,28,29,30]. Unlike their host, microorganisms have the advantage of being easily cultivable, and thus, a sustainable production of bioactive compounds can be achieved. These communities can consist of hundreds of different microbial species, making their selection and the attribution of new compounds very complex. The different issues related to microbial natural compounds production, such as strains purification and cultivation or improvement of production yields, make it difficult to efficiently operate the full potential of these microorganisms [31,32].

Therefore, it becomes necessary to set up fast and efficient methods allowing the prioritization of strains for the research and selection of bioactive compounds. Traditional selection techniques are often based on the literature, chemical or genomic profiling, bioactivity tests or bioinformatics [33,34,35,36,37,38,39]. Genomic studies allow the identification of gene clusters coding for the biosynthesis of specific specialized metabolites [33,34,35,36,37]. Metabolic profiling, which is the measurement and interpretation of the endogenous metabolite profile, investigates the changes induced by external stimuli or enhances the knowledge of inherent biological variation within subpopulations [40]. These are undoubtedly key steps to evaluate the production potential of the isolated strains [38,39,40,41]. The new bioinformatic tools belong to an interdisciplinary research field that develops methods and software tools to understand biological complex data. As an interdisciplinary field of science, bioinformatics combines biology, chemistry, physics, computer science, information engineering, mathematics and statistics. Bioinformatics is used for in silico analyses of biological queries, specifically in the field of protein affinity and activity, greatly improving the knowledge of biosynthetic pathways. Nevertheless, to date, only a few studies have combined the different approaches [25,26,42] to meet a set of objectives. However, the combination of metabolomic tools, biological tests and bioinformatic sciences may highly increases the chances of discovering molecules with structural originality and new biological targeted activities.

In this article, a methodology allowing the selection of microorganisms of interest isolated from the microbiota of *Scopalina hapalia* (ML-263), a sponge of the south-western zone of the Indian Ocean, was carried out. For this study, 9 microbial strains were pre-selected among 124 isolated from this sponge based on their potential to produce anti-aging compounds in previous biological tests [24]. The selected strains also belong to families already known in the literature for their production of metabolites of interest, such as *Bacillus* [43] or actinobacteria [25,26,27,28,29,30]. Microbial compounds such as salinosporamide, staurosporine or erythromycin have proven their biological interest [44,45,46]. During this work, the strains were prioritized based on their chemical profiles coupled with their detected biological activities. The chemical potential of the microorganisms was evaluated through high-resolution LC-HRMS/MS analyses applied to extracts from microbial cultures. Subsequently, an Ion Identity Molecular Network workflow was used to process the data [47], and the annotation of the metabolites produced was achieved using three complementary computational approaches such as the Global Natural Product Social Molecular Networking platform (GNPS) [48], the SIRIUS 5.5.7 pipeline [49,50], the ISDB-DNP tool (In Silico DataBase-Dictionary of Natural Products) [51] and the timaR package for ISDB-DNP results refinement [52]. The biological potential range of the microbial extracts was further evaluated through cytotoxic and antiplasmodial activity tests. Among the strains studied, the actinobacteria *Micromonospora fluostatini* SH-82 showed high chemical diversity, as well as valuable biological activities. This work allowed the establishment of a rigorous methodology for strain selection based on the combination of chemical and biological potentials coupled with bioinformatic tools.

## 2. Materials and Methods

### 2.1. Biological Material

#### 2.1.1. Sponge

A sponge sample of *Scopalina hapalia* (ML-263) was collected in 2013 at a 2–10 m depth. This sample was gathered on the south-west coast of Mayotte, an island located in the Indian Ocean (Kani point, GPS coordinates 12°57.624’ S; 45°04.697’ E). The sponge identification was carried out by Nicole de Voogd from the Naturalis Biodiversity Center (Leiden, The Netherlands).

#### 2.1.2. Microbial Strains

One hundred twenty-four (124) microbial strains were previously isolated and identified from *Scopalina hapalia* (ML-263) [24] by the society Genoscreen. From this microbial collection, 9 strains presented below were selected for the study (Table 1).

### 2.2. Strains Culture and Extracts Preparation

Solid cultures were carried out to obtain crude microbial extracts. The strains selected for the study were kept in storage cryotubes placed in a freezer at −80 °C. The cryoprotectant contained 10% (*v/v*) glycerol (Carlo Erba Val de Reuil, France), 10% (*w/v*) skimmed milk (Lait écrémé en poudre, Régilait, Macon, France) and 33 g/L sea salt (Instant Ocean 16 kg, Aquarium system, Sarrebourg, France). The strains were revivified by plating 100 µL of the cryotube on A1BFe+C agar medium (10 g soluble starch (ref. 417587, BD Difco, Le Pont de Claix, France), 2 g peptone (ref. 211820, BD Bacto, Le Pont de Claix, France), 4 g yeast extract (ref. 212750, BD Bacto, Le Pont de Claix, France), 33 g sea salts, 1 g CaCO_3_ (ref. 433185, Carlo Erba, Val de Reuil, France), 100 mg KBr (ref. 470735, Carlo Erba, Val de Reuil, France), 40 mg Fe_2_(SO_4_)_3_ (ref. 451926, Carlo Erba, Val de Reuil, France), 20 g agar (ref. 281210, BD Difco, Le Pont de Claix, France) to obtain 1 L of medium. The 9 cm diameter Petri dishes, (Nunc Thermo Fisher, Waltham, USA) were then incubated for 7–15 days at 28 °C in a MIR-154 PE thermostatic oven (PHC, Etten-Leur, the Netherlands).

The microbial content of two revivification Petri plates was transferred to 50 mL of sterile artificial water (Sea salts 33 g/L). Ten milliliter (10 mL) of the bacterial suspension was introduced into 100 mL of specific A1BFe+C liquid medium. This preculture was incubated at 28 °C for 7 days in a thermostated incubator at 180 rpm agitation speed (ref. S-000121948, Infors FT, Bottmingen, Switzerland).

The solid culture was made from 25 mL of this preculture. This volume was mixed with 25 g of sterile XAD-16 amberlite (ref. MFCD00145831, Sigma Aldrich, St. Louis, MI, USA) and spread on a culture Petri dish 25 × 25 cm (ref. 240835, Nunc Thermo Fisher, Waltham, MA, USA) containing 250 mL of A1BFe+C agar medium. The Petri dishes were incubated at 28 °C for 14 days.

After incubation, the resin and biomass were recovered by Buchner filtration (paper filter Whatman^®^ grade 4, 110 mm in diameter, Sigma Aldrich, St. Louis, MI, USA) and washed with demineralized water. After drying, the mixture was extracted with 100 mL of ethyl acetate (EtOAc) (ref. 448252 RPE grade, CarloErba, Val de Reuil, France) for 2 h. Following evaporation, the crude microbial extracts were used for chemical and biological analyses.

### 2.3. Chemical Analysis

#### 2.3.1. HPLC-DAD-CAD Analysis

The dry extracts were resolubilized in 100% acetonitrile (ACN) (analytical grade 99% purity, CarloErba, Val de Reuil, France) and filtered on a 0.2 μm Minisart RC filter (ref. 7764ACK, Sigma Aldrich, St. Louis, MI, USA). The samples were standardized at a concentration of 10 mg/mL and analyzed by high-performance liquid chromatography (HPLC) Dionex Ultimate 3000 (Thermo Scientific, Waltham, MA, USA) coupled to a diode array UV detector (DAD) (Thermo Scientific, Waltham, MA, USA) (190–800 nm) and a charged aerosol detector (CAD) Corona ultra RS (Thermo Scientific, Waltham, MA, USA). A Phenomenex Gemini C18 analytical column (150 × 4.6 mm, 3 µm) (Phenomenex, Torrance, CA, USA) was used for elution. A gradient solvent system with ACN (analytical grade 99% purity, CarloErba, Val de Reuil, France) (phase A) and milli-Q water (phase B), each of them containing 0.1% formic acid (FA) (analytical grade 99% purity, CarloErba, Val de Reuil, France), was used for the analyses. The extract was eluted by a linear gradient from 5 to 100% B for 30 min at a flow rate of 0.7 mL/min. The CAD signal intensity from the peaks is expressed in pA and provides quantitative data. This allows the characterization of visible (height >10 pA) or major (height > 40) peaks in Table 2.

#### 2.3.2. UHPLC-QTOF-MS/MS Analysis

The high-resolution tandem mass spectrometry (HRMS/MS) analyses were carried out by the regional platform MALLABAR at Institut Méditerranéen de Biodiversité et d’Ecologie marine et continentale IMBE (Marseille, France). The samples were dissolved in 1 mL of methanol (LCMS grade, CarloErba, Val de Reuil, France) and filtered with 0.22 µm PTFE syringe filters (Restek, Bellefonte, PA, USA). The HRMS/MS analysis was performed on a Dionex Ultimate 3000 UHPLC system (Thermo Scientific, Waltham, MA, USA) coupled to a QtoF Bruker Impact II mass spectrometer (Bruker, Billerica, MA, USA). For the separation, a Phenomenex Kinetex phenyl hexyl column (1.7 µm, 150 × 2.1 mm) (Phenomenex, Torrance, CA, USA) was used for elution. A gradient solvent system with ACN (MS grade, CarloErba, Val de Reuil, France) (phase A) and Milli-Q water (phase B), each of them containing 0.1% formic acid (FA) (analytical grade 99% purity, CarloErba, Val de Reuil, France), was used for the analyses. The extract was eluted by a linear gradient from 0 to 100% B for 8 min at a flow rate of 0.5 mL/min. Data were acquired in positive mode (ESI^+^; 20–40 eV). The main MS data acquisition parameters were: the MS1 spectra acquisition range from 20 to 1200 Da, the collision energy was set to 40 eV and the acquisition speed was 4 Hz. The 5 major precursors in MS1 were selected for the recording of the MS2 spectra.

### 2.4. Raw Data Processing

The raw data files were converted to the .mzXML format using the MS-convert software [53], which belongs to the ProteoWizard 3.0 package (Palo Alto, CA, USA). High-resolution MS data were processed using MZmine 3 software [54]. The main parameters were: a detection threshold for MS1 masses of 1^E^3 and 1^E^0 for MS2; the ADAP chromatogram builder [55] was used with a minimum scan group size of 3, a threshold of group intensity of 1^E^3, a minimum intensity of 1^E^3 and an *m/z* tolerance of 0.005 Da. The parameters used for the local minimum resolver were: a chromatographic threshold of 90%, a minimum search range of 0.05 min and a matching of MS2 scans with an *m/z* tolerance of 0.0080 Da and a retention time tolerance of 0.2 min. Feature alignment was performed with an *m/z* tolerance of 0.005 Da and a retention time of 0.08 min. The identification of the ions was carried out using the Pearson correlation coefficient as a correlation measure. An *m/z* tolerance of 0.008 Da was set for adduct detection. The full processing parameters applied are presented in the analysis batch (Appendix A). The aligned feature table and the MS1/MS2 mass spectra data file (.mgf) were exported and used for molecular network creation and compounds annotation.

### 2.5. Ion Identity Molecular Network

An Ion Identity Molecular Network (IIMN) [47] was created with the feature-based molecular networking workflow on the GNPS website [47,48,56]. The main GNPS parameters were set as follows: a precursor ion mass tolerance (PIMT) of 0.02 Da, a fragment ion mass tolerance (FIMT) of 0.02 Da and a minimum of 12 fragment ions in common. The Ion Identity Molecular Network was visualized using Cytoscape version 3.9.1 [57], and a graphic style was set to make the network more readable. A unique color was set for each strain, the size of the nodes was set according to the intensity of the precursor ions and the thickness of the bonds were set according to the cosine score (threshold: 0.7).

### 2.6. Feature Annotations

Feature annotation was performed by combining results from different computational and automated pipelines now broadly used in the metabolomic field, followed by a systematic manual inspection of the outputs to refine feature annotation. The GNPS platform enables the comparison of the MS2 spectra to several experimental spectral libraries. Only the results corresponding to a high spectral similarity score (cosine score > 0.7) were retained to ensure the highest confidence in the results. This score represents a mathematical measure of spectral similarity and is based primarily on the number of fragments in common between the MS2 spectrum recorded and the reference spectrum of the annotated compound [48]. GNPS similarity scores are given in Table 3 when the reference spectra were accessible. In parallel, MS1 and MS2 spectra were processed through the SIRIUS 5.5.7 pipeline [49] (Lehrstuhl für Bioinformatik, Jena, Germany) to calculate the feature raw formula, to predict the fragmentation pattern with CSI: fingerID module [50], and to infer the chemical class with the CANOPUS module [58]. MS2 spectra were also compared to the in silico Spectral Databases of Natural Products (ISDB) [51], and the results were refine using the timaR 2.7.2 package [52] and LOTUS database [59]. These two last tools provide a similarity score, which is also based on the similarity of fragmentation properties between the acquired MS2 spectra and those of the databases [50]. Additionally, the timaR score is also a function of the taxonomic link [52]. These different scores are displayed in Table 3. The annotation results originating from GNPS, SIRIUS and timaR were compared and are mentioned in Table 3, if they were consistent. It should be noted that for the different features detected, it is not always possible to obtain a score in each annotation pipeline, as appears in Table 3 (similarity score column). A very good confidence was assessed when each of the scores displayed elevated values on each of their independent scoring scales. Discrepant results were inspected manually and compared with other feature annotations along the molecular network to find a consensus annotation. The results were refined by inferring the fragmentation pattern from MS2 spectra. The IIMN and Network Annotation Propagation (NAP) tool [60] enabled the use of the MolNetEnhancer workflow [61] of the GNPS platform. It allowed us to propagate the chemical superclasses obtained by Classyfire [62], an open access platform permitting to identify the chemical classes of molecules into a new network, which was completed by the consensus annotations.

### 2.7. Biological Activity Tests

The biological activity tests were carried out on all the microbial extracts. The target activities were the cytotoxic activity of the HCT-116 and MDA-MB-231 cell lines and the antiplasmodial activity of *Plasmodium falciparum* 3D7. These tests were carried out by the partners of the European FEDER PHAR project, whose aim is to develop bioactive molecules from plants, marine invertebrates and microorganisms from the Indian Ocean region.

#### 2.7.1. Cytotoxic Activity

The cytotoxic activity was performed by Institut de Chimie des Substances Naturelles (ICSN, Paris, France) according to the following protocol. Human cancer cell lines were obtained from the American type Culture Collection (ATCC, Rockville, MD, USA) and were cultured according to the supplier’s instructions. Human HCT-116 colorectal carcinoma cells (ATCC^®^-CCL-247^TM^) were grown in Gibco medium RPMI 1640 (ref. 61870, Fisher Scientific, Waltham, Ma, USA) supplemented with 10% fetal bovin serum (FCS) (ref. 10500-064) and 1% glutamine. MDA-MB-231 breast carcinoma cells (ATCC^®^-HTB-26^TM^) were grown in Gibco medium DMEM (ref. 11966, Fisher Scientific, Waltham, Ma, USA) containing 4.5 g/L glucose supplemented with 10% FCS and 1% glutamine. Cell lines were maintained at 37 °C in a humidified atmosphere containing 5% CO_2_. Cell viability was determined by a luminescent assay according to the manufacturer’s instructions (Promega, Madison, WI, USA). Briefly, the cells were seeded in 96-well plates (2.5 × 10^3^ cells/well) containing 90 μL of growth medium. After 24 h of culturing, the cells were treated with the tested compounds at 1 and 10 mg/mL final concentrations. Control cells were treated with the vehicle.

After 72 h of incubation, 100 μL of CellTiter Glo Reagent (ref. G9243, Promega, Madison, WI, USA) was added for 15 min before recording luminescence, using a spectrophotometric plate reader PolarStar Omega (BMG LabTech, Champigny-sur-Marne, France). The percent viability index was calculated from three experiments.

#### 2.7.2. Antiplasmodial Activity

The in vitro culture of *Plasmodium falciparum* chloroquine-sensitive 3D7 strain (originally isolated from a patient living in the Netherlands) was performed following the Trager and Jensen procedure [63]. The host cells used were human red blood cells (A+), and the culture medium was composed of RPMI 1640 (Gibco, Fisher Scientific, Waltham, MA, USA) containing NaHCO_3_ (32 mM), HEPES (25 mM) and L-glutamine. It was supplemented with 1.76 g/L of glucose (Sigma-Aldrich), 44 mg/mL of hypoxanthine (Sigma-Aldrich), 100 mg/L of gentamycin (Gibco, Fisher Scientific, Waltham, MA, USA) and 10% human pooled serum (A+), as described in this study [64]. The strain was obtained from ATCC. The crude extract solutions were prepared in DMSO and tested in a series of dilutions in a 96-well plate, with the highest concentration of DMSO not causing any toxicity to the parasite. The parasite’s growth was measured after 48 h of incubation by determining the lactate dehydrogenase activity. The positive control for all the experiments was artemisinin (analytical standard, Sigma-Aldrich, Saint-Quentin-Fallavier, France). The IC_50_ values were calculated from the resulting graph (GraphPad Prism, Ritme, Paris, France).

## 3. Results

### 3.1. Metabolomic Study

The objective of this work was to select the most promising microorganisms from a strains library according to their chemical profile and biological screening.

#### 3.1.1. HPLC-DAD-CAD Analysis

HPLC-DAD-CAD analyses were performed on all the microbial extracts standardized at a concentration of 10 mg/mL and then, they were compared to a culture medium blank. This blank corresponds to the extract from medium A1 with amberlite without microbial inoculation. HPLC-CAD provided universal detection and allowed us to evaluate the relative amount of microbial metabolites present in the extracts. HPLC-DAD analyses also provided complementary information for compounds displaying characteristic UV-chromophores. Appendix A shows the details of visible peaks (height > 10 pA) and those considered as major ones (>40 pA) that may correspond to these specialized metabolites for each strain studied on HPLC-CAD chromatograms. The analyses indicate that several peaks also present in the blank can be detected in the microbial extracts. It is not surprising that in addition to the microbial compounds, amberlite also captures some compounds from the medium. Therefore, the peaks similar to those of the blank were not taken into account in the analysis. Table 2 summarizes the peaks exclusively observed for each microbial extract.

Two strains presented more than 10 specific visible peaks, *Chaetomium globosum* SH-123 (10 peaks) and *Micromonospora fluostatini* SH-82 (17 peaks). From the latter, eight are considered as major ones. Appendix A presents an example of an HPLC-CAD chromatogram for the strain *Micromonospora fluostatin*i SH-82. These experiments provide a quantitative profile of the extracts and make it possible to assess future difficulties in isolating the compounds.

To rank and select promising microorganisms, a “chemical score” was then attributed to the strains. For the HPLC-CAD analysis, the score was determined according to the number of visible and major peaks, which is detailed in Section 3.3 on strain selection.

#### 3.1.2. Ion Identity Molecular Network

In order to evaluate and explore, in more detail, the chemical composition of the extracts, an Ion Identity Molecular Network (IIMN) [47] was created using the GNPS platform [48]. This network allows the visualization and an easy comparison of the chemical diversity of all the microbial extracts analyzed. This graphic representation is composed of numerous nodes representing precursor ions and cleaned of the signals coming from the culture medium. As ESI generates many in-source mono- and multi-charged adducts, the IIMN allows the discrimination of adducts (ion identity node) and their combination into a single node (collapsed node) corresponding to the same chemical entity. It thus helps to reduce the size and complexity of the networks. The final network shown in Figure 1 includes 812 nodes, 47% of which are grouped into 61 clusters (>2 nodes), with the rest being single nodes (feature nodes). In total, 57 collapsed nodes were identified.

This network makes it easy to observe the distribution and specialization of microbial metabolites, each strain being represented by a distinct color. The shape of the nodes represents: the grouping of several adducts into a single node (cluster node) and the identification of a single adduct (ion identity node), such as [M+H]^+^ or nodes whose adduct could not be identified (feature node). Their size is proportional to the precursor ion intensity. Most of the nodes in the network are unique to each species, and only some of them are common to two or more strains. These nodes are grouped in the box on the bottom right side, and despite the annotation effort, they did not allow the identification of remarkable compounds. The framework at the top right represents the *Bacillus paralicheniformis* SH-02 and *B. licheniformis* SH-68 extracts. It consists of the largest cluster composed of 99 nodes, 75% of which are common to both strains. There are six large clusters (>10 nodes) specific to one species or to both *Bacillus* strains. Two of these clusters come from *Salinispora arenicola* SH-78′s extract. Figure 2 shows the number of strain-specific nodes and the ones common to several strains.

*Bacillus licheniformis* SH-68′s extract exhibits the highest number of nodes (235), and thus, a high chemical richness, but most of them are shared with other strains, indicating a rather low chemical specificity (43). *Salinispora arenicola* SH-78, *Micromonospora fluostatini* SH-82 and *Micromonospora citrea* SH-89 have a high number of unique nodes, with 134, 130 and 101, respectively. In *Micromonospora fluostatini* SH-82′s extract, this represents 72% of its total nodes’ number, highlighting a great metabolome specialization. For the Ion Identity Molecular Network, the “chemical score” was given as a function of the number of total and unique nodes. The scoring method is explained further.

Node dereplication was performed by combining the results from different computational and automated pipelines, GNPS [48], SIRIUS [49,50] and timaR [51,52]. In total 11 nodes were annotated via GNPS, 57 nodes via SIRIUS and 27 nodes with timaR. Twenty-three (23) nodes were annotated with good confidence, as they were present in at least two annotation pipelines. These annotations, presented in Table 3, allowed the putative identification of known metabolites, associated chemical classes and possible nodes of interest. This work completed the chemical superclass network (Appendix A) obtained through the MolNetEnhancer workflow [61].

This additional network (Appendix A) shows a diversity of superclasses according to Classyfire [62], such as oxygenated organic compounds or phenylpropanoids/polyketides, within the various crude extracts. The main superclass is the organoheterocyclic compounds, representing 57% of the nodes, most of them coming from the large clusters of *Bacillus licheniformis* SH-02 and *Bacillus paralicheniformis* SH-68. In this cluster, five nodes could be annotated by GNPS as tryptamines or derivatives and confirmed by SIRIUS. A diversity of chemical superclasses was observed within the same species, notably in two microorganisms, *Micromonospora fluostatini* SH-82 and *Salinispora arenicola* SH-78. An annotation task was carried out on the main clusters for each strain. Three microbial extracts display a significant number of relevant annotations. Twenty, fourteen and thirteen nodes were annotated in an accurate manner from *Micromonospora fluostatini* SH-82, *Chaetomium globosum* SH-123 and *Salinispora arenicola* SH-78 respectively, and their clusters are detailed below. For this last strain, the two main clusters could not be clearly annotated, which could indicate the possible presence of new metabolites or of compounds still unlisted in the databases.

#### 3.1.3. Focus on *Micromonospora fluostatini* SH-82

The focus was on the *Micromonospora fluostatini* SH-82 strain (red node) because of the number and size of the clusters, the diversity of superclasses and the ratio of unique nodes. The main annotated clusters are shown in Figure 3. The shape and the size are described as in the previous network and the thickness of the edge represents the cosine score (spectral similarity). An annotation is proposed for every node of the network, but with different levels of specificity. The molecule’s structure was reported when an assignation was possible (black frame and blue frame for isomers). Otherwise the molecular formula was calculated (red frame) and it corresponds to analogs for which it was not possible to retrieve the exact structure. This can potentially indicate the existence of new compounds.

All the annotations made on the molecular network are presented in Table 3. If possible, for each of them, a SIRIUS, GNPS and ISDB score were mentioned, in addition to other information such as the *m/z*, the raw formula or the compound name. The larger cluster 3.a is composed of 32 nodes, 37.5% of which are annotated. On the right side, annotated nodes (black frame) belonging to the megalomicins can be observed. Five megalomicins could be identified with a high degree of accuracy (SIRIUS score > 80%). During the analyses, these molecules formed doubly charged ions, which are not displayed on this cluster. Appendix A presents a zoomed-in depiction of these nodes, where a concordance in the masses of the doubly charged ions and their very high intensity are more noticeable compared to the rest of the cluster. These nodes were not detected by the SIRIUS software, highlighting the importance of paying attention to the loss of information related to the data processing parameters used. In the left part of cluster 3.a, the nodes have been annotated as belonging to different erythromycins or derivatives. For erythromycin C (*m/z* 720.4529 [M+H]^+^, C_36_H_65_NO_13_) and erythromycin D (*m/z* 704.4586 [M+H]^+^, C_36_H_65_NO_12_), the confidence of the similarity score is very high, at, respectively, 95.82% and 93.53%. The presence of three isomers (blue box) of erythromycin C with [M-H_2_O+H]^+^ adducts and *m/z* 702.442 Da is also observed. The annotated nodes have been described by CANOPUS as the erythromycin class of natural products. These annotations have chemical similarities, which reinforce the coherence of all our identifications. Figure 3b shows different clusters composed of nodes annotated as erythronolides or derivatives. In this figure, the identification is less accurate, with a SIRIUS score between 60 and 75%. The collapsed node described as 6-deoxyerythronolide B (*m/z* 369.2624 [M-H_2_O+H]^+^ C_21_H_38_O_6_) gathers four nodes representing different adducts such as [M-2H_2_O+H]^+^ or [2M+Na]^+^. This type of node avoids redundancy and enhances the annotation by identifying adducts, and thus, the neutral mass. The MS1 spectrum analysis of this collapsed node is presented in Appendix A, justifying the adducts detected and leading to the SIRIUS annotation of the compound. The unidentified nodes (red frame) have masses and raw formula similar to those of the neighboring annotated nodes. They can therefore describe isomers in the case of megalomicins or new molecules of potential interest. These nodes could not be precisely identified by the SIRIUS software, the ISDB tool or GNPS platform, but a raw formula has been proposed.

#### 3.1.4. Focus on *Chaetomium globosum* SH-123 and *Salinispora arenicola* SH-78

A focus on these two strains that showed a large number of annotations is presented below.

The *Chaetomium globosum* SH-123 cluster (Figure 4a) consists of 13 nodes and was annotated (Table 3). The nodes were annotated as belonging to chaetoglobosins or derivatives. Sixty-four percent (64%) of the annotations presented a SIRIUS similarity score above 80%. Chaetoglobosin C (*m/z* 529.2699 [M+H]^+^, C_32_H_36_N_2_O_5_) and chaetoglobosin A (*m/z* 529.2693 [M+H]^+^, C_32_H_36_N_2_O_5_) had high scores of 91.01% and 97.67%, respectively. Both of these nodes are isomers, and the SIRIUS software made several high-scoring proposals. To accurately identify these metabolites, isolation and NMR analysis would be required. The four prochaetoglobosins were annotated with a SIRIUS score between 64% and 82% and have a high spectral similarity score (cosine score > 0.9). This score indicates close chemical structures, and therefore, reinforces the proposed annotations.

Figure 4b shows the clusters from *Salinispora arenicola* SH-78, and the details of the annotations are presented in Table 3. Three compounds have been identified by GNPS as stauroporin (*m/z* 467.2085 [M+H]^+^, C_28_H_26_N_4_O_3_), 7-OH-staurosporin (*m/z* 483.2028 [M+H]^+^, C_28_H_26_N_4_O_4_) and rifamycin S (*m/z* 696.3022 [M+H]^+^, C_37_H_45_NO_12_). Two other annotated clusters show structures similar to that of rifamycin S, which may represent isomers of proansamycin B (*m/z* 624.31 [M+H]^+^, C_34_H_41_NO_10_), 34a-deoxy-rifamycin W (*m/z* 640.3119 [M+H]^+^, C_35_H_45_NO_10_) or 25-deacetoxy-25-hydroxyrifamycin S (*m/z* 654.2918 [M+H]^+^, C_35_H_43_NO_11_). The crude extract of *Salinispora arenicola* SH-78 allowed annotations of various structures such as saliniketals, rifamycins or staurosporin. The two main clusters (Appendix A) could not be clearly annotated by the software used, indicating possible novelties from this microorganism.

All the accurately annotated nodes gathered from the molecular networks are summarized in Table 3. It includes complementary information, such as the *m/z* with the associated adduct, the molecular formula, the compound name or the corresponding chemical family. The similarity score presented was obtained using the different annotation tools, GNPS, SIRIUS and timaR. The maximum similarity score values are respectively 100%, 1 and 1, when indicating a perfect spectral similarity between the spectrum from the acquired data and the reference spectrum. The superscript letters and numbers indicate which annotation tools the information came from. For some nodes, all three tools led to the same result, thus reinforcing the annotation obtained. The workflow used therefore allowed to assess the chemical potential of each micro-organism, putatively identify some known metabolites and put us on the trail of new compounds.

### 3.2. Biological Activity

All nine microbial extracts have been tested for their cytotoxic and antiplasmodial activities. These activities were the biological targets of the European FEDER PHAR project, and the tests were carried out by our partners and the co-authors.

#### 3.2.1. Cytotoxic Activity

The cytotoxicity of the extracts was evaluated by measuring the viability of the HCT-116 (colon carcinoma) and MDA-MB-231 (breast cancer) cell lines, two models currently used in therapeutic research in cancer [65,66]. Figure 5 present the percentage of viability of the HCT-116 (Figure 5a) and MDA-MB-231 (Figure 5b) cell lines in the presence of the extracts at two different concentrations, 10 μg/mL and 1 μg/mL. The activity is named as “promising” when the percentage of viability is lower than 50% at a concentration of 1 µg/mL.

Among all the microbial extracts, three extracts showed promising activity against the HCT-116 cell line, the extracts *Bacillus berkeleyi* SH-137, *Chaetomium globosum* SH-123 and *Salinispora arenicola* SH-78, with viability percentages at 1 µg/mL of 50 ± 0.1%, 37 ± 0.5% and 4 ± 0.1%, respectively. The extract of *Micromonospora fluostatini* SH-82 exhibited moderate activity with a viability of 19 ± 3% at a concentration of 10 µg/mL, but this percentage increases sharply at a concentration of 1 µg/mL.

There is a similarity between the histograms reporting the extracts’ cytotoxicity against the two cell lines. Only the extract from *Salinispora arenicola* SH-78 showed promising activity against the MDA-MB-231 cell line, with a percentage of viability of 12 ± 1% at 1 µg/mL. The extracts from *Chaetomium globosum* SH-123 and *Micromonospora chokoriensis* SH-36 gave rise to very similar activities, with percentages of 52 ± 3% and 50 ± 0.7% at 1 µg/mL respectively. These activity tests attest that the extracts of *Chaetomium globosum* SH-123 and *Salinispora arenicola* SH-78 possess notable cytotoxic activities for both tests. The extract from *Salinispora arenicola* SH-78, with percentages of viability lower than 15% for the two cell lines, is particularly promising.

#### 3.2.2. Antiplasmodial Activity

The second activity type targeted was antiplasmodial activity. Figure 6 shows the percentages of inhibition of the microbial extracts against the *Plasmodium falciparum* 3D7 strain at two different concentrations, 50 µg/mL and 10 µg/mL. The activity was considered to be noteworthy, as the percentage of inhibition was at least 50% at a concentration of 10 µg/mL.

All the extracts exhibit more than 50% inhibition at a concentration of 50 µg/mL, therefore, the activity was evaluated at a lower concentration (10 µg/mL). Among these extracts, the results considered as promising were obtained with *Bacillus berkeleyi* SH-137, *Chaetomium globosum* SH-123, *Micromonospora fluostatini* SH-82 and *Salinispora arenicola* SH-78, which showed inhibition percentages exceeding 50% at the lowest concentration of 10 µg/mL. The inhibition rates reached 52%, 70%, 71% and 71% respectively. This preliminary analysis makes it possible to quickly screen the extracts of interest before carrying out a second more precise analysis.

The concentration required to inhibit the growth of the parasite culture by 50% (IC_50_) was measured for these four pre-selected extracts to assess their antiplasmodial potential (Figure 7). The extracts were considered to be promising with an IC_50_ below 15 µg/mL and very promising with an IC_50_ below 5 µg/mL.

Consequently, the extracts from *Chaetomium globosum* SH-123 and *Micromonospora fluostatini* SH-82 are described as promising, with IC_50_ values of 5.4 ± 2.5 and 11.3 ± 1.2 µg/mL respectively. *Salinispora arenicola* SH-78′s extract exhibited very promising antiplasmodial activity, with an IC_50_ of 2.6 ± 0.9 µg/mL.

Among the nine strains studied, several of them showed interesting biological activities. The two extracts from *Chaetomium globosum* SH-123 and *Salinispora arenicola* SH-78 demonstrated remarkable biological activities. They possess both cytotoxic activity against the two cell lines tested, as well as antiplasmodial activity. *Bacillus berkeleyi* SH-137′s extract has a promising cytotoxic activity, and *Micromonospora fluostatini* SH-82′s extract showed interesting antiplasmodial effects. As with the chemical screening, a biological score was assigned to each strain in order to prioritize the microbes according to the number and type of biological activity. Details are presented in Section 3.3.

### 3.3. Strain Selection

The objective of this study was to identify, from a library of bacterial strains, those that might be of interest for the isolation of new molecules for potential therapeutic applications. To prioritize these promising isolates, we performed a preliminary chemical and biological screening on the cultures’ crude extracts. For each analysis, a score was assigned to the strains, the details of which are presented in Table 4. These scores were used to evaluate and rank the potential of the microorganisms.

The chemical score (CS) is the combination of the HPLC-DAD-CAD and IIMN scores. The first components (CS_VP_ and CS_MP_) are based on the number of visible and major peaks detected on the HPLC-CAD chromatograms. The IIMN scores (CS_UN_ and CS_TN_) were attributed according to the number of unique and total nodes present in the molecular network. The final chemical score is calculated out of 10, with the formula below, Equation (1):CS = (CS_VP_ + CS_MP_ + CS_TN_ + CS_UN_)/2(1)

For example, for the crude extract of *Micromonospora fluostatini* SH-82, a large number of major peaks (17 peaks) and unique nodes (130 nodes) were observed, attributing the maximum score for each analysis and a total chemical score of 10/10.

The biological score (BS) was assigned according to the intensity of the targeted biological activities. This score was calculated by adding the average of the cytotoxic activity scores towards the two cell lines (BS_CA,HCT-116_ and BS_CA,MDA-MB-231_) to the antiplasmodial activity score (BS_AA_) in order to obtain a score out of 10 according to the formula below, Equation (2):BS = (BS_CA, HCT-116_ + BS_CA, MDA-MB-231_)/2 + BS_AA_(2)

For example, *Salinispora arenicola* SH-78′s extract showed an inhibition percentage >80% for cytotoxicity, and thus, obtained an average of 5/5. Its antiplasmodial score, which was 5/5 because of its inhibition percentage > 70%, was added to the previous calculated mean to obtain a total biological score of 10/10 for this strain.

In order to obtain a higher probability of chemical novelty, a “novelty” score (NS) out of 3 was also defined. It was increasingly assigned according to the decreasing number of metabolites already described from the species in the LOTUS [59], DNP [67] and NP atlas [68] databases. In the case of *Micromonospora fluostatini* SH-82, no literature references were found, which gave it the maximum novelty score of 3.

The average of these three scores gives the final selection score calculated as follows with Equation (3), with maximum possible score 10.
Selection score = (CS + BS + NS)/2.3(3)

It allowed us to rank the microorganisms according to their overall potential. Detailed scoring for each microbial extract is presented in Appendix A. Table 5 synthesizes the ranking of the nine strains studied based on these different scores.

Three strains have a selection score below four, indicating their low potential under these culture conditions and for the targeted biological activities. The first three strains in the ranking have a selection score above 6.5, indicating a real potential. *Chaetomium globosum* SH-123 (6.5), *Salinispora arenicola* SH-78 (7.8) and *Micromonospora fluostatini* SH-82 (8.5) demonstrated good results in the biological and chemical screening. The literature and databases showed that the first two strains mentioned are already known to be producers of metabolites of interest, and therefore support the approach taken in this study. *Micromonospora fluostatini* SH-82, which ranked first with a selection score of 8.5, has not been studied much so far. This microorganism can therefore be attributed a strong potential as a candidate for the search of new bioactive metabolites based on the selected criteria.

## 4. Discussion

In response to many public health problems, researchers are pursuing research on the chemistry of natural substances to find new therapeutic pathways [9,10,32]. Marine macroorganisms such as sponges are already known to produce metabolites with structural originality and remarkable biological activities [11,12]. Within these hosts, there are microbiota that can consist of hundreds of bacteria belonging to different genera [22,23,24]. These bacteria may be responsible for the production of specialized metabolites in holobionts and exhibit a great potential for the discovery of new molecules with diverse biological activities [23,24,25,26,27,28,29,30]. However, the large number of microorganisms isolated from these complex systems makes their exploration fastidious, time-consuming and expensive for research. In addition, many microbial metabolites have already been discovered, which can lead to dead ends in new studies. The scientific community is therefore looking for ways to prioritize the microorganisms of interest among the huge number of strains [33,34,36,37,38,39]. This study proposes a methodology to select the most promising strains isolated from the microbiota in Indian Ocean sponge, *Scopalina hapalia* (ML-263), in regard with their chemical and biological potential.

The first part of the chemical screening was carried out through HPLC-DAD-CAD analyses of all the crude extracts. These experiments made it possible to evaluate the richness of specialized metabolites and allowed the initial scoring. The strains produced compounds of different polarities, with retention times ranging from 9 to 45 min. *Micromonospora fluostatini* SH-82 showed the highest number of high-intensity peaks with different retention times. The results obtained indicated the presence of many compounds directly coming from the culture medium in the microbial extracts. Further experiments should focus on reducing this presence or enriching the extracts in metabolites of interest, using amberlite for instance [69]. However, this quantitative analysis helps to guide the future isolation, purification and identification work.

The previous experiments were supplemented by studying the high-resolution data obtained from HRMS/MS analyses. HRMS/MS data allowed the creation of an Ion Identity Molecular Network (IIMN) [47] and of a second network describing the chemical superclass [60,61] of the different compounds. Based on the dereplication of microbial metabolites carried out using the different annotation tools (GNPS platform [48], SIRIUS software [49,50] and ISDB database coupled to timaR [51,52]), this work allowed to evaluate the metabolites production potential according to the number of unique nodes detected joined to the annotations performed. The IIMNs (Ion Identity Molecular Networks, Figure 1) and the molecular network describing the chemical classes (Appendix A) highlight the considerable diversity of metabolites produced by the panel of microorganisms isolated from the sponge *Scopalina hapalia* (ML-263). The work of Cheng et al. (2015) [25] on sponges also demonstrated the presence of microorganisms able to produce bioactive metabolites, supporting the interest in these macroorganisms. The present IIMNs consist of 812 nodes, the majority of which are unique to each microbial species (Figure 2), indicating a high degree of specificity, except in the case of *Bacillus spp.* From our study, *Micromonospora fluostatini* SH-82 and *Salinispora arenicola* SH-78 are the most remarkable strains because of their large number of specific nodes, indicating a greater possibility of species-specific metabolites. Searching the LOTUS [59], DNP [67] and NPatlas [68] databases confirms this hypothesis for the *Salinispora arenicola* SH-78 strain, with an average of 44 already known metabolites. Oppositely, for *Micromonospora fluostatini* SH-82, none of the numerous compounds detected have already been described, thus reinforcing its chemical interest. The exploited workflow produced nodes with identifiable adducts, which provided the neutral mass, and then, a more detailed annotation [47]. Thus, a complete annotation task was performed on the molecular network, with a special focus in this article, on the most important clusters.

The fungus *Chaetomium globosum* SH-123 has already been extensively studied, with at least 294 molecules described in the LOTUS database [59]. The chaetoglobosins annotated in our molecular network are confirmed by the literature [70,71]. The low number of nodes (45) obtained from this extract compared to the number of metabolites already described indicates a low chance of discovering novelties in the selected culture conditions. Studies have shown that the use of other media, such as PDA (Potatoe Dextrose Agar) or sterilized moistened rice medium, allows this microorganism to produce a higher diversity of interesting metabolites [70,71].

*Salinispora arenicola* SH-78 has been described in many studies as a model microorganism for the production of innovative metabolites [72]. Annotations performed on the crude extract match with those in the literature, as we observed the possible presence of several known molecules such as staurosporine or its derivatives. For this latter one, the three bioinformatic tools used in this study led to the same results, reinforcing the identifications achieved. METACYC^®^ data [73] and the study by Stramann et al. (2016) highlighted the link between the biosynthetic pathways of rifamycins and saliniketals [74]. In the present study, we have potentially identified some of these metabolites, such as rifamycin S, saliniketals, 34a-deoxy-rifamycin W or 25-*O*-deacetyl-27-*O*-demethylrifamycin S. Despite the annotation tasks carried out, the different tools applied and the extensive literature gathered, the two main clusters of this strain obtained in our results (Appendix A) could not be clearly identified. It would therefore be interesting to initiate the isolation and identification of these compounds due to the existing potential of this genus and the diversity of bioactive compounds it can produce [72].

The third strain that showed relevant annotations is *Micromonospora fluostatini* SH-82. Numerous unique nodes annotated belonging to a bioactive family of erythromycins were detected. These include the megalomicins, macrolide antibiotics isolated from a soil bacterium, *Micromonospora megalomicea* [75]. Useglio et al. (2010) described the in vivo bioconversion of erythromycin C to megalomicin A and identified the different precursors and genes implicated [76]. METACYC ^®^ database [73] recognized the different precursors involved in this biosynthetic pathway, such as erythromycin C, erythromycin D, 3-*O-*alpha mycarosyl erythronolide B, erythronolide B, and finally, 6-deoxyerythronolide B [76,77]. Thanks to the different annotation tools, in our study, all these molecules were annotated from the crude extract of *Micromonospora fluostatini* SH-82. The concordance of the information obtained, such as the presence of the compounds in the biosynthetic pathway, the consensus annotations and the source species, reinforce the identification of this chemical family from our extract. The numerous unknown nodes in the megalomicin cluster or its precursors could, therefore, correspond to unknown derivatives and could be of real interest from a chemical point of view.

In the case of *Bacillus licheniformis* SH-68 and *B. paralicheniformis* SH-02, accurate annotations have been made, showing the possible presence of tryptamine in the crude extracts. One study showed the ability of *Bacillus cereus* to produce tryptamines in a medium containing tryptophan [78].

For the other strains, few precise annotations have been achieved. Additional work is necessary by researchers directly working on the MS spectra to obtain a stable crude formula.

Two of the selected criteria allowing the prioritization of the strains were the biological activities of the crude extracts. The cytotoxicity was tested towards the HCT-116 and MDA-MB-231 cell lines, respectively cell models for colon and breast cancers [65,66]. Two extracts thus showed considerable activity. The first one, from *Salinispora arenicola* SH-78, has very high cytotoxic activity levels with percentages of viability of 4 ± 0.1% and 12 ± 1% at a concentration of 1 µg/mL for the HCT-116 and MDA-MB-231 lines, respectively. The second one from *Chaetomium globosum* SH-123 showed 37 ± 0.2% and 52 ± 3% viability in the cytotoxicity tests at the same concentration. The dereplication enabled the of the possible presence of compounds from the chaetoglobosin family, such as chaetoglobosin A, chaetoglobosin C or 20-dihydrochaetoglobosin A. The study of Li et al. (2014) revealed the activity of several chaetoglobosins from the same species *Chaetomium globosum* towards the HCT-116 cell line. Chaetoglobosin A and 20-dihydrochaetoglobosin A showed stronger cytotoxicity activity with IC_50s_ of 3.15 and 8.44 µM compared to that of etoposide as a control, which had an IC_50_ of 2.13µM [71]. Huang et al. (2016) [79] showed the activity of other chaetoglobosins against the MDA-MB-435 cell line also using a model for breast cancer control [80]. This work [79] demonstrated that chaetoglobosin C and A have IC_50s_ values of 19.97 and 37.56 µM, respectively, against this biological target. These studies and the review published by Chen et al. (2020) [70] indicated that this family of molecules possesses a diversity of biological activities (antifungal, antitumor, antibiotic, etc. [70]), which may explain the results obtained for the crude extract studied. For *Salinispora arenicola* SH-78′s extract, dereplication also allowed the potential identification of some compounds confirmed in the literature. In the study of Jimenez et al. (2012) [44], staurosporine, which was used as a control, and a mixture of OH-staurosporine showed IC_50s_ values of 58.24 nM and 83.83 nM, respectively, against the HCT-8 cell line, a model used in the fight against colon cancer. The same molecules also showed activity against the MDA-MB-435 cell line with IC5_0s_ values of 28.68 and 215.42 nM. The work by Xiao et al. (2018) showed the significant cytotoxicity activity of this type of molecules against the HCT-116 cell line [80]. *Salinispora arenicola* has an exceptional chemical diversity, with many metabolites having biological activities such as rifamycin B [74] or cyclomarins [81].

For the screening of antiplasmodial activity against the *Plasmodium falciparum* 3D7 strain, three extracts emerged, two of which already showed promising cytotoxic activities and include molecules with multiple biological actions. Schulze et al. (2015) [82], describing 11 salinipostins from *Salinispora sp.*, demonstrated remarkable activities against malaria parasites. The other notable extract comes from *Micromonospora fluostatini* SH-82. The main metabolites annotated were from the erythromycin family, especially the megalomicins. The study by Goodman et al. (2012) showed the antiplasmodial activity of megalomicin against *Plasmodium falciparum* 3D7, and also, against an azitromycin-resistant strain of the parasite [46]. The presence of a large number of megalomicins and derivatives could explain the IC_50_ of 11.3 ± 1.2 µg/mL of the crude extract. The activities described as promising are of real interest, given the low concentration of secondary metabolites in the crude extracts. It would be very interesting to target and isolate the pure compounds responsible for these activities.

Additionally, a “novelty” scoring was established for each species according to the number of metabolites already known from the databases. This type of score is in line with other pipelines aimed at assigning chemical novelty to extracts [83]. This scoring made it possible to identify strains that have already been extensively studied. Three strains exhibited more than forty known metabolites, including one *Chaetomium globosum* isolate with more than 200 references. *Salinispora arenicola* has been studied in great detail, with the authors describing its metabolome under different culture conditions [84]. The objective of this study was, therefore, to create a selection methodology to prioritize the most promising strains inside a strain library. This step is indeed crucial when one is confronted with a large number of microorganisms [34]. A current method is the genomic characterization of microbial strains. This approach allows researchers to anticipate the metabolites produced through the identification of genes involved in biosynthetic pathways of interest. This way is very promising because it makes it possible to evaluate the potential of the strain, even before extract production. It can also be applied to a very large number of microorganisms [36,37]. However, this approach has certain limitations, as some genes are silent in certain culture conditions and may not be expressed during the strain’s cultivation [85]. This technique requires specific materials and does not necessarily allow the quantification of the targeted metabolite produced. The study of the metabolome of the microbial strains is also an effective means of selection [41] that can be associated with biological screening [86].

The method described in this study may be difficult to transpose to hundreds of bacteria due to the cultivation technique and the extracts’ production. However, it is possible to extend it to a large panel of microbes by adapting the cultivation process. Indeed, Ortlieb et al. [87] demonstrated the possibility to perform actinomycete culturing in microplates. This can be used for a rapid chemical and biological screening through numerous microbial strains. The use of microbial extracts produced in microplates could incredibly accelerate the detection process. However, the further upscaling from microplates cultures to larger productions (Petri dishes, Erlenmeyers or bioreactors) sometimes encounters problem [88], which is not the case in our study. Our prioritization method could additionally be coupled with the OSMAC [89] approach to evaluate the potential of each microorganism, and also, the most favorable growing conditions for the production of bioactive metabolites.

## 5. Conclusions

The objective of the study was to set up a method enabling the prioritization inside a panel of microorganisms isolated from an Indian Ocean sponge, *Scopalina hapalia* (ML-263). The aim was to identify the most promising strains for the production of bioactive metabolites as cytotoxic or antiplasmodial compounds and perform a preliminary step before the isolation and the characterization of the molecules. For this method, a chemical screening was carried out based on the realization of an Ion Identity Molecular Network (IIMN) and a quantitative evaluation of the microbial raw extracts’ composition. Among the nine strains studied, three of them, *Micromonospora fluostatini* SH-82, *Salinispora arenicola* SH-78 and *Chaetomium globosum* SH-123, presented numerous relevant annotations obtained through different bioinformatic tools (SIRIUS, GNPS and ISDB). Staurosporins, rifamycins, megalomicins and chaetoglobosins have been annotated and are described as families of molecules with various biological activities. In parallel with this chemical screening, the raw extracts were evaluated for cytotoxicity against HCT-116 and MDA-MB-231 cell lines, as well as for antiplasmodial activity against the *Plasmodium falciparum* 3D7. *Chaetomium globosum* SH-123 and *Salinispora arenicola* SH-78 showed remarkable activity on both cancer lines. In addition, a third isolate, *Micromonospora fluostatini* SH-82, showed promising antiplasmodial activity. A scoring process was set up for each analysis carried out on the microbial extracts studied. The overall ranking of the strains took into account the chemical and the biological scores, as well as a “novelty” score depending on the number of metabolites already known from the species. *Micromonospora fluostatini* SH-82 was ranked at number one for its novelty, chemical diversity and promising antiplasmodial activity. The final classification of our microbial isolates shows the relevance of our selection method. Indeed, among the top three, two microorganisms *Salinispora arenicola* and *Chaetomium globosum* have already shown their remarkable potential for bioactive metabolites production. The results obtained, and thus the assigned scores, highly depend on the applied culture conditions, which play an important role on the metabolome produced by the microbial strains. However, this selection method deserves to be applied to larger collections of microorganisms or even to other organisms such as plants or marine macroorganisms by adapting the scoring to the nature of the objectives and results.

## Figures and Tables

**Figure 1 microorganisms-11-00697-f001:**
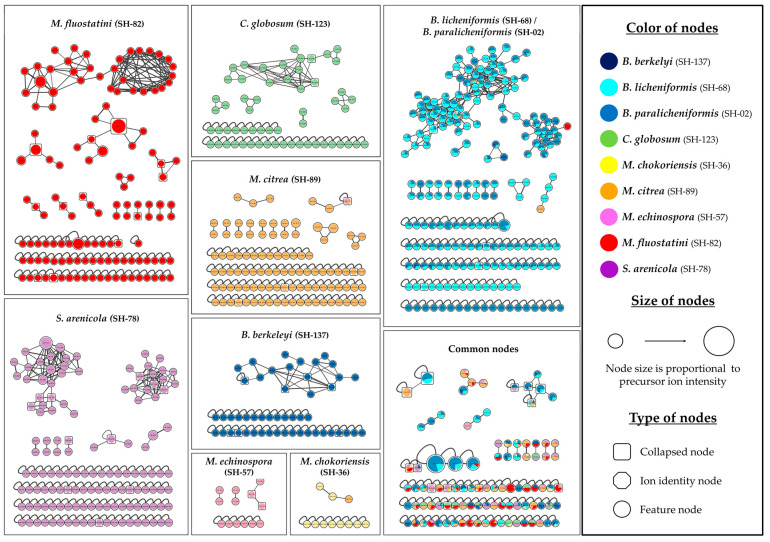
Ion Identity Molecular Network (IIMN) of the 9 microbial extracts. Node’s color describes the bacterial species, node’s shape is related to the adduct type and the collapsed property and node’s size is proportional to precursor ion intensity (The figure is in high definition, with the possibility of zooming-in).

**Figure 2 microorganisms-11-00697-f002:**
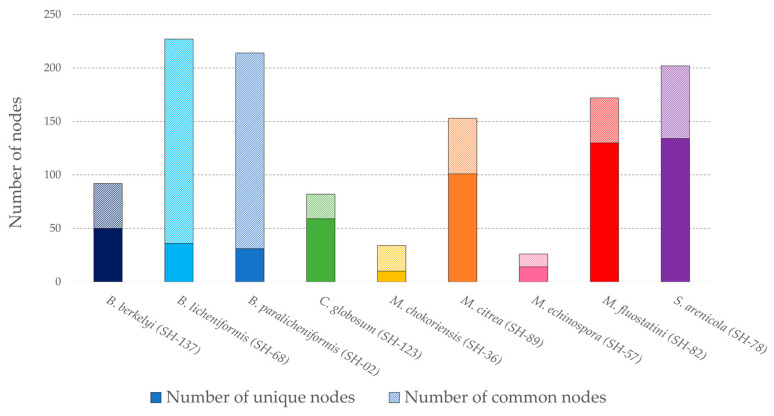
Histogram representing the number of unique (plain pattern) and common (striped patterns) nodes for each strain.

**Figure 3 microorganisms-11-00697-f003:**
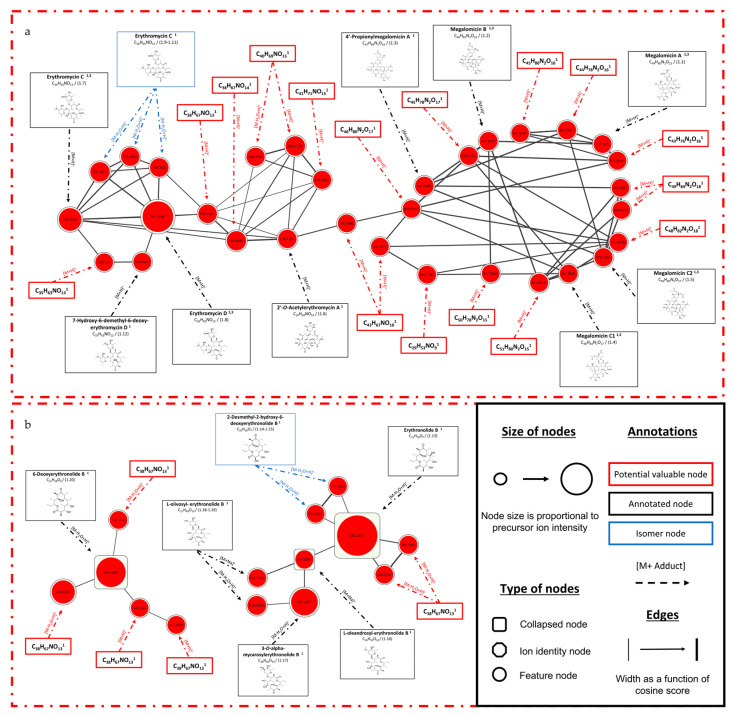
Main annotated clusters from the crude extract of *Micromonospora fluostatini* SH-82 (red node). (**a**) Main cluster representing the natural product class of erythromycins and (**b**) set of clusters representing the erythronolides family and derivatives. Superscript numbers in the annotations correspond to data from ^1^ SIRIUS, ^2^ GNPS or ^3^ ISDB timaR bioinformatics tools. (The figure is in high definition, with the possibility of zooming-in).

**Figure 4 microorganisms-11-00697-f004:**
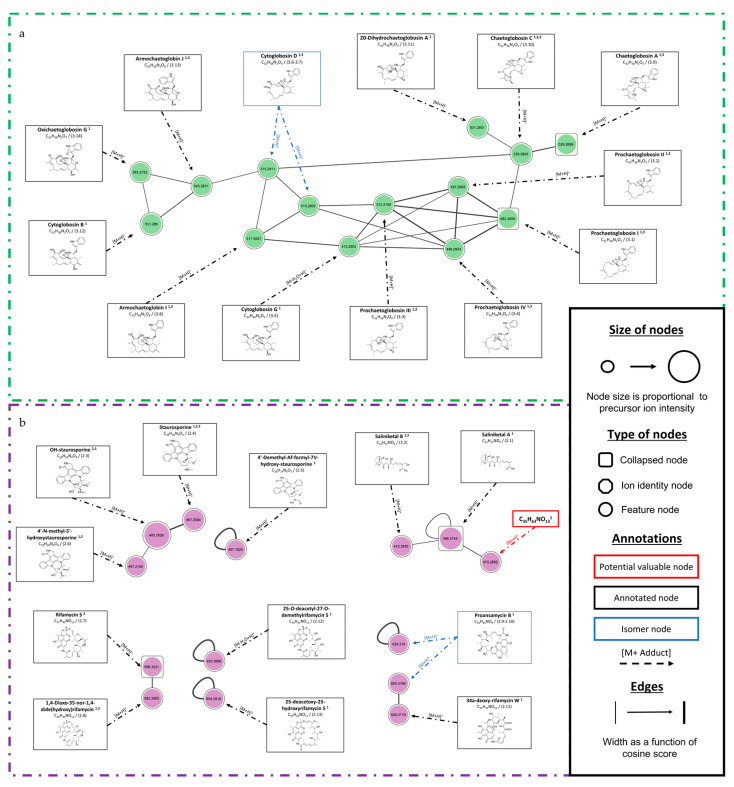
Annotated clusters from the microbial extracts of (**a**) *Chaetomium globosum* SH-123 and (**b**) *Salinispora arenicola* SH-78. Superscript numbers in the annotations correspond to data from ^1^ SIRIUS, ^2^ GNPS or ^3^ ISDB timaR bioinformatics tools. (Figure is in high definition, with the possibility of zooming-in).

**Figure 5 microorganisms-11-00697-f005:**
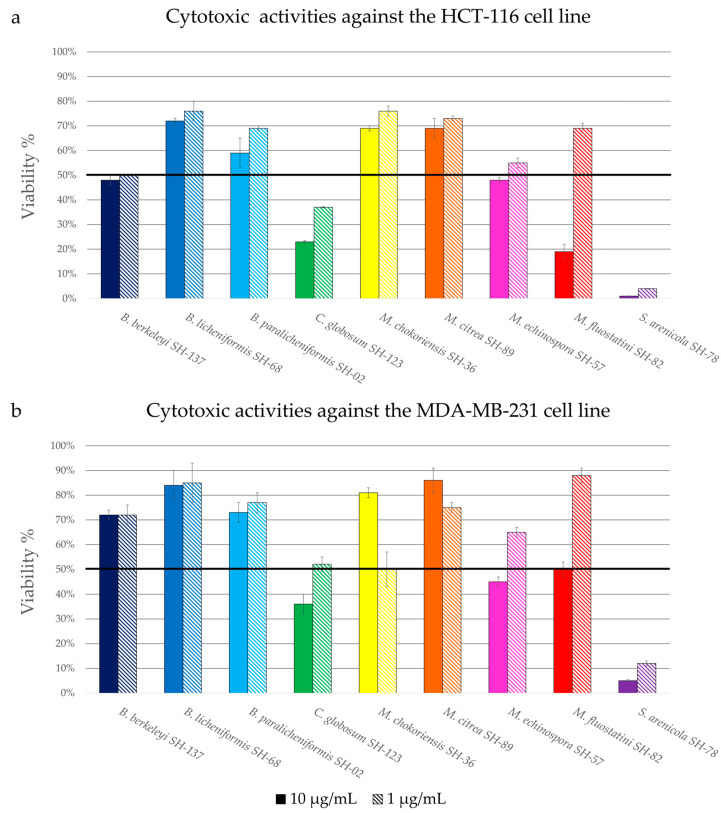
Cytotoxicity of the crude microbial extracts against the HCT-116 (**a**) and the MDA-MB-231 (**b**) cell lines, showing the ratio of viability. The different colors correspond to the different microbial strains. Horizontal black line indicates the threshold for promising strains (viability < 50%).

**Figure 6 microorganisms-11-00697-f006:**
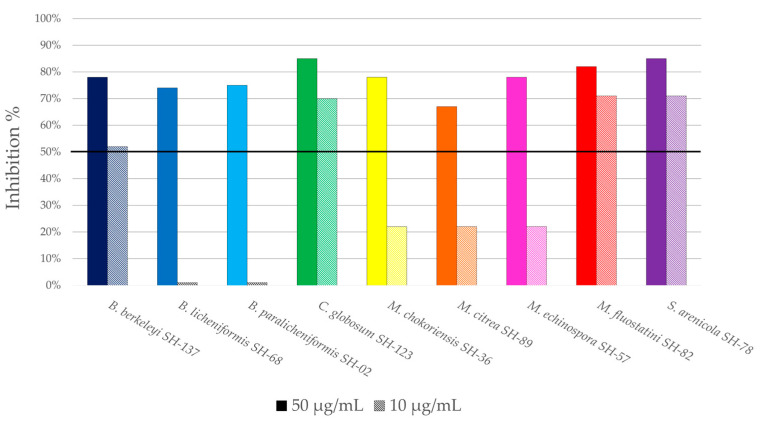
Antiplasmodial activities of crude microbial extracts against *P. falciparum* 3D7 strain, showing the ratio of inhibition at two concentrations. The different colors correspond to the different microbial strains. Horizontal black line indicates the threshold for promising strains (inhibition > 50%).

**Figure 7 microorganisms-11-00697-f007:**
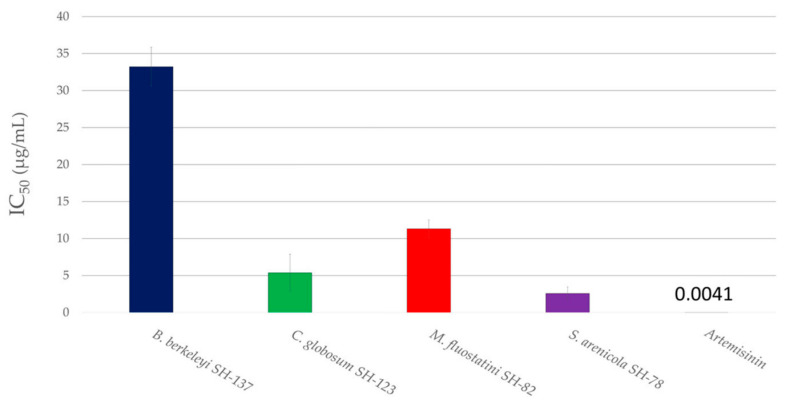
IC_50_ of the four microbial extracts of interest against *P. falciparum* 3D7 strain. The different colors correspond to the different microbial strains.

**Table 1 microorganisms-11-00697-t001:** Microbial strains isolated from *Scopalina hapalia* (ML-263) selected for the study.

Class	Species	Code	Selected Regions for Genetic Characterization
*Bacilli*	*Bacillus berkeleyi*	SH-137	
	*Bacillus paralicheniformis*	SH-02a	ADNr 16s (from V1 to V5)
	*Bacillus licheniformis*	SH-68	
*Sordariomycete*	*Chaetomium globosum*	SH-123	ITS/*BenA*
	*Micromonospora chokoriensis*	SH-36	
	*Micromonospora citrea*	SH-89	
*Actinobacteria*	*Micromonospora echinospora*	SH-57	ADNr 16s (from V1 to V5)
	*Micromonospora fluostatini*	SH-82	
	*Salinispora arenicola*	SH-78	

**Table 2 microorganisms-11-00697-t002:** Number of (visible and major) peaks observed for each microbial extract.

Microbial Strain	Visible Peaks (Height > 10 pA)	Major Peaks (Height > 40 pA)
*Bacillus berkeleyi* SH-137	6	2
*Bacillus paralicheniformis* SH-02a	6	3
*Bacillus licheniformis* SH-68	4	2
*Chaetomium globosum* SH-123	10	3
*Micromonospora chokoriensis* SH-36	0	0
*Micromonospora citrea* SH-89	1	0
*Micromonospora echinospora* SH-57	1	0
*Micromonospora fluostatini* SH-82	17	8
*Salinispora arenicola* SH-78	7	0

**Table 3 microorganisms-11-00697-t003:** Summary table of annotations from the Ion Identity Molecular Network (IIMN).

Strain	Compound ID	Retention Time (min)	*m/z* [Adduct]	Molecular Formula ^(1)^	Compound Name ^(1,2,3)^	Superclass ^(a)^/Natural Product Class ^(b)^	Similarity Score ^(1,2,3)^
*Micromonospora* *fluostatini* SH-82	1.1	6.14	877.5648 [M+H]^+^	C_44_H_80_N_2_O_15_	Megalomicin A ^(1,3)^	Organic oxygen compounds^(a)^/ Erythromycins^(b)^	92.2% ^(1)^/0.22 ^(3)^
	1.2	6.34	919.5754 [M+H]^+^	C_45_H_78_N_2_O_17_	Megalomicin B ^(1,3)^		88.61% ^(1)^/0.47 ^(3)^
	1.3	6.52	933.5939 [M+H]^+^	C_47_H_84_N_2_O_16_	4′-Propionylmegalomicin A ^(1)^		89.01% ^(1)^
	1.4	6.59	961.5883 [M+H]^+^	C_48_H_84_N_2_O_17_	Megalomicin C1 ^(1,3)^		87.19% ^(1)^/0.47 ^(3)^
	1.5	6.77	975.6046 [M+H]^+^	C_48_H_86_N_2_O_17_	Megalomicin C2 ^(1,3)^		83.55% ^(1)^/0.40 ^(3)^
	1.6	7.26	776.4797 [M+H]^+^	C_39_H_69_NO_14_	2′-*O*-Acetylerythromycin A ^(1)^		83.58% ^(1)^
	1.7	6.75	720.4529 [M+H]^+^	C_36_H_65_NO_13_	Erythromycin C ^(1,3)^		95.82% ^(1)^/0.22 ^(3)^
	1.8	7.08	704.4586 [M+H]^+^	C_36_H_65_NO_12_	Erythromycin D ^(1,3)^		93.53% ^(1)^/0.22 ^(3)^
	1.9	7.61	702.4425 [M-H_2_O+H]^+^	C_36_H_65_NO_13_	Erythromycin C ^(1)^		95.55% ^(1)^
	1.10	7.36	702.4423 [M-H_2_O+H]^+^	C_36_H_65_NO_13_			93.29% ^(1)^
	1.11	7.13	702.442 [M-H_2_O+H]^+^	C_36_H_65_NO_13_			94.40% ^(1)^
	1.12	6.98	690.4431 [M+H]^+^	C_36_H_65_NO_12_	7-Hydroxy-6-demethyl-6-deoxy-erythromycin D ^(1)^		83.46% ^(1)^
	1.13	7.54	385.2577 [M-H_2_O+H]^+^	C_21_H_38_O_7_	Erythronolide B ^(1)^	Phenylpropanoids and polyketides ^(a)^	61.07% ^(1)^
	1.14	7.30	371.2423 [M-H_2_O+H]^+^	C_20_H_36_O_7_	2-Desmethyl-2-hydroxy-6-deoxyerythronolide B ^(1)^		68.90% ^(1)^
	1.15	7.23	371.2423 [M-H_2_O+H]^+^				68.90% ^(1)^
	1.16	7.97	569.3287 [M+Na]^+^	C_28_H_50_O_10_	L-oleandrosyl-erythronolide B ^(1)^		75.17% ^(1)^
	1.17	7.99	529.3367 [M-H_2_O+H]^+^		3-*O*-alpha-mycarosylerythronolide B ^(1)^		68.21% ^(1)^
	1.18	7.71	555.3136 [M+Na]^+^	C_27_H_48_O_10_	L-olivosyl- erythronolide B ^(1)^		75.93% ^(1)^
	1.19		515.3209 [M-H_2_O+H]^+^				75.87% ^(1)^
	1.20	8.25	369.2626 [M-H_2_O+H]^+^	C_21_H_38_O_6_	6-Deoxyerythronolide B ^(1)^		63.47% ^(1)^
	1.21	2.06	136.0618 [M+H]^+^	C_5_H_5_N_5_	Adenine ^(1,3)^	Organoheterocyclic compounds ^(a)^/Purine alkaloids ^(b)^	100% ^(1)^/0.63 ^(3)^
*Salinispora arenicola* SH-78	2.1	7.84	396.2745 [M+H]^+^	C_22_H_37_NO_5_	Saliniketal A ^(1)^	Lipids and lipid-like molecules ^(a)^/Open-chain polyketides ^(b)^	49.50% ^(1)^
	2.2	6.90	412.2693 [M+H]^+^	C_22_H_37_NO_6_	Saliniketal B ^(1,3)^		55.44% ^(1)^/0.45 ^(3)^
	2.3	7.20	483.2028 [M+H]^+^	C_28_H_26_N_4_O_4_	OH-staurosporine ^(2,3)^	Organoheterocyclic compounds ^(a)^/Carbazole alkaloids ^(b)^	0.79 ^(2)^/0.30 ^(3)^
	2.4	7.33	467.2085 [M+H]^+^	C_28_H_26_N_4_O_3_	Staurosporine ^(1,2,3)^		98.11% ^(1)^/0.96 ^(2)^/0.30 ^(3)^
	2.5	7.53	497.1825 [M+H]^+^	C_28_H_24_N_4_O_5_	4′-Demethyl-Af-formyl-7V-hydroxy-staurosporine ^(1)^		81.75% ^(1)^
	2.6	7.30	497.2192 [M+H]^+^	C_29_H_28_N_4_O_4_	4′-N-methyl-5′-hydroxystaurosporine ^(1,3)^		62.56% ^(1)^/0.28 ^(3)^
	2.7	9.70	696.3022 [M+H]^+^	C_37_H_45_NO_12_	Rifamycin S ^(2,3)^	Phenylpropanoids and polyketides ^(a)^ / Ansa macrolides ^(b)^	0.71 ^(2)^/0.35 ^(3)^
	2.8	9.24	682.2866 [M+H]^+^	C_36_H_43_NO_12_	1,4-Dioxo-35-nor-1,4-dide(hydroxy)rifamycin ^(1,3)^		74.69% ^(1)^/0.39 ^(3)^
	2.9	9.39	624.3166 [M+H]^+^	C_34_H_41_NO_10_	Proansamycin B ^(1)^		43.29% ^(1)^
	2.10	8.30	624.3190 [M+H]^+^				64.34% ^(1)^
	2.11	9.75	640.3119 [M+H]^+^	C_35_H_45_NO_10_	34a-deoxy-rifamycin W ^(1)^		54.25% ^(1)^
	2.12	9.01	622.2666 [M-H_2_O+H]^+^	C_34_H_41_NO_11_	25-*O*-deacetyl-27-*O*-demethylrifamycin S ^(1)^		52.72% ^(1)^
	2.13	8.68	654.2918 [M+H]^+^	C_35_H_43_NO_11_	25-deacetoxy-25-hydroxyrifamycin S ^(1)^		71.40% ^(1)^
	2.14	6.84	176.0707 [M+H]^+^	C_10_H_9_NO_2_	4-Hydroxy-1-methyl-2-quinolone ^(1,3)^	Organoheterocyclic compounds ^(a)^ / Quinoline alkaloids ^(b)^	75.64% ^(1)^/0.54 ^(3)^
*Chaetomium globosum* SH-123	3.1	10.4	483.3009 [M+H]^+^	C_32_H_38_N_2_O_2_	Prochaetoglobosin I ^(1,3)^	Organoheterocyclic compounds ^(a)^/Cytochalasan alkaloids ^(b)^	64.04% ^(1)^/0.45 ^(3)^
	3.2	10.27	497.2805 [M+H]^+^	C_32_H_36_N_2_O_3_	Prochaetoglobosin II ^(1,3)^		69.16% ^(1)^/0.61 ^(3)^
	3.3	9.94	513.2749 [M+H]^+^	C_32_H_36_N_2_O_4_	Prochaetoglobosin III ^(1,3)^		81.75% ^(1)^/0.40 ^(3)^
	3.4	9.74	499.2953 [M+H]^+^	C_32_H_38_N_2_O_3_	Prochaetoglobosin IV ^(1,3)^		80.73% ^(1)^/0.56 ^(3)^
	3.5	9.49	515.2902 [M-H_2_O+H]^+^	C_32_H_40_N_2_O_5_	Cytoglobosin G ^(1)^		86.58% ^(1)^
	3.6	9.02	515.2903 [M+H]^+^	C_32_H_38_N_2_O_4_	Cytoglobosin D ^(1,3)^		77.72% ^(1)^/0.61 ^(3)^
	3.7	8.24	515.2911 [M+H]^+^	C_28_H_24_N_4_O_5_	4′-demethyl-Af-formyl-7V-hydroxy-staurosporine ^(1)^		84.31% ^(1)^/0.53 ^(3)^
	3.8	8.97	517.3057 [M+H]^+^	C_32_H_40_N_2_O_4_	Armochaetoglobin I ^(1,3)^		79.76% ^(1)^/0.61 ^(3)^
	3.9	9.31	529.2693 [M+H]^+^	C_32_H_36_N_2_O_5_	Chaetoglobosin A ^(1,3)^		97.67% ^(1)^/0.59 ^(3)^
	3.10	8.62	529.2699 [M+H]^+^	C_32_H_36_N_2_O_5_	Chaetoglobosin C ^(1,2,3)^		91.01% ^(1)^/0.80 ^(2)^/0.55 ^(3)^
	3.11	8.74	531.2851 [M+H]^+^	C_32_H_38_N_2_O_5_	20-Dihydrochaetoglobosin A ^(1)^		94.02% ^(1)^
	3.12	8.26	531.286 [M+H]^+^	C_32_H_38_N_2_O_5_	Cytoglobosin B ^(1)^		88.00% ^(1)^
	3.13	8.67	545.2651 [M+H]^+^	C_32_H_36_N_2_O_6_	Armochaetoglobin J ^(1,3)^		84.42% ^(1)^/0.59 ^(3)^
	3.14	8.39	563.2752 [M+H]^+^	C_32_H_38_N_2_O_7_	Oxichaetoglobosin G ^(1)^		69.84% ^(1)^
*Bacillus* *licheniformis* SH-68 / *Bacillus paralicheniformis* SH-02	4.1	10.89	385.3206 [M+H]^+^	C_25_H_40_N_2_O	N-[2-(1H-indol-3-yl)ethyl]pentadecanamide ^(1,2)^	Lipids and lipid-like molecules ^(a)^/N-acyl amines ^(b)^	97.53% ^(1)^/0.93 ^(2)^
	4.2	11.27	413.3514 [M+H]^+^	C_27_H_44_N_2_O	Heptadecanoic acid tryptamide ^(1,2)^		87.54% ^(1)^/0.93 ^(2)^
	4.3	10.74	371.3046 [M+H]^+^	C_24_H_38_N_2_O	Myristoyl tryptamine ^(1,2)^	Organoheterocyclic compounds ^(a)^/Cytochalasan alkaloids ^(b)^	92.12% ^(1)^/0.93 ^(2)^
	4.4	11.13	399.3365 [M+H]^+^	C_26_H_42_N_2_O	N-palmitoyltryptamine ^(1,2)^		100% ^(1)^/0.92 ^(2)^
	4.5	2.07	161.1070 [M+H]^+^	C_10_H_12_N_2_	Tryptamine ^(1,2,3)^	Organoheterocyclic compounds ^(a)^/Simple indole alkaloids ^(b)^	100% ^(1)^/0.77 ^(2)^/0.38 ^(3)^
	4.6	8.02	245.1646 [M+H]^+^	C_15_H_20_N_2_O	N-[2-(1H-indol-3-yl)ethyl]-2-methylbutanamide ^(2,3)^		87.03% ^(1)^/0.15 ^(3)^
	4.7	4.23	173.1072 [M+H]^+^	C_11_H_12_N_2_	Triptoline ^(1,2,3)^	Organoheterocyclic compounds ^(a)^/Carboline alkaloids ^(b)^	98.95% ^(1)^/0.77 ^(2)^/0.12 ^(3)^
*Micromonospora* *chokoriensis* SH-36	5.1	6.52	176.0703 [M+H]^+^	C_10_H_9_NO_2_	Indole-3-acetic acid ^(1,2)^	Organoheterocyclic compounds ^(a)^/Simple indole alkaloids ^(b^	100% ^(1)^/0.94 ^(2)^

Data from ^1^ SIRIUS, ^2^ GNPS or ^3^ ISDB timaR bioinformatics tools. Scores are only reported for computational tools that provide an annotation for each of the feature displayed. Each of these scores are independent and related to the specific workflow they originate from. ^a^ Super class from Classyfire; ^b^ Natural products class from CANOPUS.

**Table 4 microorganisms-11-00697-t004:** Description of the scoring methodology for each score assigned to the strains’ extracts (chemical, biological and novelty score).

Chemical score (CS) ^(1)^							
HPLC-DAD-CAD Analysis				Ion Identity Molecular Network (IIMN) Analysis			
Number of Visible Peaks (VP)	Score VP	Number of Major Peaks (MP)	Score MP	Number of Total	Score TN	Number of Unique Nodes (UN)	Score UN
	(CS_VP_)		(CS_MP_)	Nodes (TN)	(CS_TN_)		(CS_UN_)
0–2	1	0–1	1	0–40	1	0–30	1
3–5	2	2–3	2	40–80	2	30–60	2
6–8	3	4–5	3	80–120	3	60–90	3
9–11	4	6–7	4	120–160	4	90–120	4
>11	5	>7	5	>160	5	>120	5



**Biological Score (BS) ^(2)^**				**Novelty Score (NS)**			

**Cytotoxic Activity**	**CA Score**	**Antiplasmodial Activity (AA)**	**AA Score**	**Average of Bibliographic Reference**		**Score (NS)**	
**(CA)**	**(BS_CA_)**		**(BS_AA_)**				
>80%	5	>70%	5	>50		0	
80–50%	4	70–50%	4	30–50		1	
50–40%	3	50–40%	3	10–30		2	
40–30%	2	40–30%	2	<10		3	
<30%	1	<30%	1				
>80%	5	>70%	5				
80–50%	4	70–50%	4				
50–40%	3	50–40%	3				

^(1) ^Chemical score, CS = (CS_VP_ + CS_MP_ + CS_TN_ + CS_UN_)/2. ^(2)^ Biological score, BS = (BS_CA, HCT-116_ + BS_CA, MDA-MB-231_)/2 + BS_AA._

**Table 5 microorganisms-11-00697-t005:** Synthesis of the scores attributed to each of the 9 microbial strains and general ranking based on chemical, biological and novelty scores.

Strain	Biological Score	Chemical Score	Novelty Score	Selection Score	Ranking
*Bacillus berkeleyi* SH-137	6.5	5	3	6.3	4
*Bacillus paralicheniformis* SH-02	2.5	6	3	5.0	5
*Bacillus licheniformis* SH-68	1.5	5	1	3.3	8
*Chaetomium globosum* SH-123	9	6	0	6.5	3
*Micromonospora chokoriensis* SH-36	3.5	2	3	3.7	7
*Micromonospora citrea* SH-89	1.5	5	3	4.6	6
*Micromonospora echinospora* SH-57	3.5	2	1	3.3	8
*Micromonospora fluostatini* SH-82	6.5	10	3	8.5	1
*Salinispora arenicola* SH-78	10	7	1	7.8	2

## Data Availability

The molecular networks produced for this study are available with these IDs: Ion Identity Molecular Networking: ID=47c2fbaeac0445aca72d04dc31ade11b. MolnetEnhancer network: ID=a057946bb91e42d6b959062d13c1db2.

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
