# Peer review of "Prioritization of Microorganisms Isolated from the Indian Ocean Sponge Scopalina hapalia Based on Metabolomic Diversity and Biological Activity for the Discovery of Natural Products"

_microorganisms, 2023, doi:10.3390/microorganisms11030697_

Round 1

Reviewer 1 Report

This manuscript reports the screening of a collection of bacterial species isolated from a single species of sponge, for cytotoxicity and antiplasmodial actitivty.  The most promising of these bacteria as sources for potentially useful drugs were ranked by these criteria. The paper is well written but more details are required to allow the reader to understand the key table 3.

The word “anticancer” has been used inappropriately (26 times) throughout the manuscript (and in the keywords) to describe the effects of these compounds on two cell lines. It is quite wrong to use this term. These drugs were not used to treat cancers, they were merely tested on two “cancer” cell lines. No “normal” cell lines were used as a comparison and to justify the term “anticancer” at all, many transformed cell lines need to be compared to many non-transformed cell lines and even this would not really test anticancer activity since cancer is a disease of the entire animal. I suggest that the term should be substituted for “cytotoxicity” in all cases. The single test for cell viability /cytotoxicity was for ATP production. The line 36 in the abstract “showed remarkable anticancer … activity” is absolutely unwarranted.

Table 3 requires a much more complete description. The legend should define RT and reveal what is meant by similarity? What is similar to what and why are two values x/x sometimes given?

Figure 5. Are the error bars SE or SD and were these repeated experiments or just repeated wells?

Figure 6 should show error bars?

Line 131. Reference [1] is not correct. It seems that no reference is required here?

Line 288 The full name “Plasmodium falciparum” should be given here.

Line 473. “rifamycins” not “ryfamicins”?

References 19, 24, 43, 78 and probably others have given capital to species names (the original papers did not) and this is incorrect (e.g. Bacillus Subtilis instead of Bacillus subtilis)

Author Response

Thank you.

Reviewer 2 Report

The novelty and the quality of the manuscript are good and it does not need extensive improvement before publication. It is carefully organized and written. It is easy to follow it and contains clear comments and conclusions.  In my opinion, this manuscript is very detailed and meticulous, it covers all the literature in the field with critical point of view. The topic have been completely covered and is well connected through the text. There is a significant  novelty in presented topic.  For all these reasons, I can recommend the acception of the manuscript after minor revision. Only small suggestions to emphasized.

 1. The superiority of  HPLC-CAD than HPLC-DAD method should be more emphasized.

 2. Not all of the described results are covered in the discussion section.

 3. No all information was given of  activity of isolates.

Author Response

Thank you.
